# On the Quality and Validity of Course Evaluation Questionnaires Used in Tertiary Education in Greece

Ilias Papadogiannis [1,*], Costas Vassilakis [2], Manolis Wallace [1] and Athanassios Katsis [3]

1 ΓAB LAB—Knowledge and Uncertainty Research Laboratory, University of the Peloponnese, Akadimaikou G. K. Vlachou, 22131 Tripoli, Greece; wallace@uop.gr
2 Department of Informatics and Telecommunications, University of the Peloponnese, Akadimaikou G. K. Vlachou, 22131 Tripoli, Greece; costas@uop.gr
3 Department of Social and Educational Policy, University of Peloponnese, Damaskinou & Kolokotroni, 20131 Korinthos, Greece; katsis@uop.gr
* Correspondence: i.papadogiannis@uop.gr

**Abstract:** In compliance with national legislation, Greek tertiary education institutions assess educational quality often using a standardized anonymous questionnaire completed by students. This questionnaire aims to independently evaluate various course components, including content organization, instructor quality, facilities, infrastructure, and grading methods. Despite widespread use across universities, the questionnaire's validity remains unexamined. This study addresses this gap by analyzing 48,000+ questionnaire responses from the University of the Peloponnese (2014–2022), encompassing 68 undergraduate and graduate programs. Confirmatory factor analyses were used to assess the quality of the questionnaire, while exploratory factor analyses were used to assess the dimensions of the tool based on the data. Both analyses reveal shortcomings: confirmatory analysis detects strong correlations between supposedly different factors, and exploratory analysis identifies dimensions inconsistent with the expected structure. These findings question the questionnaire's quality and the validity of drawn conclusions, while additionally identifying opportunities for reducing the number of questions, which can contribute to increased questionnaire submission rates. Given its common use across Greek universities and its influence on shaping courses, urgent redesigning of the questionnaire for tertiary education evaluation is recommended.

**Keywords:** quality assurance; higher education; CFA; EFA

## 1. Introduction

Quality assurance in higher education is a self-evident goal for institutions, as its internationalization has led to an increasing call for accountability and the need to develop a culture of quality in order to meet the challenges of globalized higher education [1]. It refers to a systematic, organized, and ongoing commitment to quality. It implies the establishment of an internal system of principles, standards, and norms, the proper functioning of which is confirmed by periodic internal and external evaluation methods [2]. The major goal of implementing quality assurance methods throughout Europe was to instill trust in the quality of educational outcomes, provide assurance that academic standards are being protected and improved, and provide a good return on public investment in higher education [3].

An important aspect of an internal evaluation system is the evaluation of courses by students. Course evaluation serves as an important process through which higher education institutions receive feedback from students regarding courses [4]. Typically conducted anonymously at the end of a semester, student evaluations allow universities to assess how positively a course was perceived, the competency of teaching, and the materials provided, and to gain insights into potential areas for improvement. Analysis of student evaluations of courses can provide important information for institutional management

and instructors on how to improve the student experience [5,6]. For students, it provides an opportunity to directly share their perspectives on which elements of a course were effective or which may have been lacking. By thoughtfully and honestly compiling student evaluation data across campuses, institutional leaders can identify areas for improvement and implement changes that increase student satisfaction and success [6].

There is a debate about the validity of student evaluations of teaching (SETs) in higher education [7]. Some studies have supported the validity of SETs, while others have challenged it [8–10]. Literature reviews that have examined this issue have been inconclusive [11,12]. Spooren et al. [11] found that validity was affected by the methods used, and in many cases, tools developed by the organizations themselves were used rather than standardized scientific tools. In addition, scholars argue that SET ratings reflect students' assessments of teachers as persons rather than the quality of teaching [11], and this view is also supported by complementary studies [12,13]. Even students' assessments of course outcomes appeared to be biased [14].

More specifically, student evaluations of teaching (SETs) have long been criticized and debated as a tool for evaluating instructional quality and effectiveness. Research has shown that SETs can lead to unacceptably high error rates and misclassify teachers' performance [15]. Furthermore, SETs have little correlation with learning, and groups such as women faculty and faculty of color tend to be disadvantaged in SET ratings [16]. Additional studies have found that students reward lenient grading and easy courses with higher SETs, and instructors feel pressure to achieve good SET ratings [17]. Other studies have also demonstrated that biases related to race, gender, discipline, and other factors can negatively impact minority and female instructors' SET scores, even when course design and content are held constant [18]. Together, this body of research suggests that SETs alone are an imperfect and potentially biased measure of teaching effectiveness that should be supplemented with multiple qualitative and quantitative measures in evaluation processes. Given the mixed results and lack of consensus, the validity of SETs remains an open question.

In Greece, the process of student evaluation of courses has been a reality for several decades. The main tool is the "Student Course Evaluation Questionnaire", a tool recommended by the Hellenic Quality Assurance and Accreditation Agency (HQA) (https://www.ethaae.gr/en, accessed on 12 January 2024). This tool was developed in 2007 and is used either without any adaptations or with minor modifications by the institutions (a draft here). This paper attempts to evaluate the structure of the questionnaire using data from the University of the Peloponnese, a regional university in the country, in order to assess whether the measures of the questionnaire are consistent with the understanding resulting from the separation of its individual dimensions. These dimensions are evident from the separation of the questions into categories, as shown in the questionnaire.

This study contributes to the evaluation of the institutions because demonstrating the existence of specific factors allows university administrations to examine different aspects of teaching. In this way, universities can examine the scores on the various factors to identify strengths and areas for improvement. Ensuring that measurement tools have a strong evidence base is critical for the purposes of quality assurance and fair assessment of teaching.

## 2. Data and Methods

The data used in this study originate from the internal evaluation process of the University of the Peloponnese and, more specifically, from the students' evaluation forms regarding various factors of the educational work. The questionnaires are distributed during the instruction period, between the 8th and 10th weeks of teaching, and are completed anonymously by the students. It is emphasized that the "questionnaires" are used only for quality assurance purposes within the institution, including the preparation of annual internal evaluation reports and the improvement of the teaching process and infrastructure, and they are not used for the evaluation of teaching staff.

The dataset utilized in this study included information about the department, the course, the academic year, and assessment data on the following aspects: the course, supporting teaching (if applicable), assignments (if applicable), teaching staff, the course laboratory (if applicable), and the student's self-assessment. The type of each general variable is shown in Table 1.

**Table 1.** Variable types.

| Variable | Type |
| --- | --- |
| Department_Id | Numeric |
| Department_Name | Text |
| Course_Title | Text |
| Course_Code | Alphanumeric |
| Course_Year | Numeric |
| Course_Code | Alphanumeric |
| Course_Category | Numeric |
| Evaluation questions (1–37) | Numeric |

Course evaluation variables were provided on a five-point Likert-type scale (1 = unacceptable, 2 = unsatisfactory, 3 = moderate, 4 = satisfactory, and 5 = very good). The variable 'Course category' indicated whether the course instruction included laboratory practice or not. The questionnaire statements were divided into sections, corresponding to the dimensions factors, as shown in Table 2.

**Table 2.** Questionnaire items per factor.

| Course | Supporting/Assistive Teaching | Assignments | Teaching Staff | Lab | Self-Assessment |
| --- | --- | --- | --- | --- | --- |
| 1. The objectives and requirements of the course were comprehensible. | 11. Usefulness of supporting/assistive teaching. | 15. Were home assignments given in a timely fashion? | 22. Does the instructor organize the presentation of the material in the lectures well? | 29. How would you rate the laboratory difficulty level compared to the year it is taught? | 33. I attend the lectures regularly. |
| 2. The course content corresponded to the objectives of the course. | 12. Rate the quality of supporting/assistive teaching. | 16. Were the assignment delivery or presentation deadlines reasonable? | 23. The instructor inspires interest/enthusiasm for the course subject. | 30. Are the notes/handouts adequate regarding the laboratory exercises? | 34. I attend the labs regularly. |
| 3. Each class was clearly structured and organized. | | 17. Was there relevant research literature available in the library? | 24. The instructor analyses and presents concepts in a straightforward and interesting manner using examples. | 31. Are the basic principles of the experiments/exercises well explained? | 35. I respond to written assignments/exercises consistently. |
| 4. The educational material used aided towards the understanding of the subject. | | 18. Was appropriate guidance provided by the instructor? | 25. The instructor encourages students to ask questions and to develop their critical ability. | 32. Is the laboratory equipment/ infrastructure adequate? | 36. I study the course material systematically. |
| 5. Were the educational aids (textbooks, notes, additional bibliography) provided in a timely fashion? | | 19. Were the instructor's comments constructive and detailed? | 26. The instructor was punctual in his/her obligations (attendance in class, timely correction of assignments or lab reports, student office hours)? | | 37. I dedicate the following number of hours on a weekly basis to study the particular course: |
| 6. How satisfactory was the textbook(s) and/or the notes? | | 20. Was the opportunity to improve the assignment provided? | 27. The instructor develops a spirit of collaboration with all students. | | $1 \leq 2$ h, 2 = 2–4 h, 3 = 4–6 h, 4 = 6–8 h, $5 \geq 8$ h |

**Table 2.** *Cont.*

| Course | Supporting/Assistive Teaching | Assignments | Teaching Staff | Lab | Self-Assessment |
|---|---|---|---|---|---|
| 7. Was the bibliography easily accessible in the institution's library? | | 21. Did the given assignment(s) help you to better understand the subjects/topics? | 28. The teaching assistant(s) is (are) helpful in better understanding the course content and/or fieldwork. | | |
| 8. Do you consider the course prerequisites necessary? | | | | | |
| 9. Use of knowledge from other courses/linking to other courses? | | | | | |
| 10. How would you rate the difficulty of the course compared to the year it is taught? | | | | | |
| 13. How would you rate the number of ECTS units compared to the workload? | | | | | |
| 14. Transparency of grading criteria. | | | | | |

The dataset included 48.008 questionnaire submissions for eight years, from the academic year 2014–2015 to 2021–2022. The questionnaires span across 68 undergraduate and postgraduate programs and 2380 courses. Due to the different types and requirements of the courses, there were missing entries in some variables, mainly in questions related to supporting teaching, labs, and assignments.

### 2.1. Data Preparation

As noted above, there were missing values in some questions concerning supporting teaching, assignments, and the course's laboratories. This was a systematic characteristic of the questionnaire and indeed a desired outcome, since students should not be expected to rate aspects of the course that are not actually present (e.g., labs for courses not involving laboratory practice). For this reason, the dataset was divided into eight subsets according to whether the above dimensions were present or not. In this way, eight new subsets of data were created (Table 3). After splitting into subsets, only missing entries in var8 were mentioned in the prerequisites for the course. The missing values were filled in with the median of the lesson.

**Table 3.** Data subsets according to the courses' structures and requirements.

| Subset No. | Laboratory | Assignments | Supporting Teaching | No of Records |
|---|---|---|---|---|
| 1 | √ | √ | √ | 4274 |
| 2 | √ | √ | x | 3464 |
| 3 | √ | x | √ | 948 |
| 4 | √ | x | x | 2259 |
| 5 | x | √ | √ | 15,910 |
| 6 | x | √ | x | 9725 |
| 7 | x | x | √ | 4033 |
| 8 | x | x | x | 7395 |
| | | Total | | 48,008 |

√ indicates presence of answers in dataset, x indicates absence.

*2.2. Methods*

To assess the fit of the model, a series of confirmatory factor analyses (CFAs) were first conducted on eight sub-datasets. In other words, CFA was used to determine whether the data are consistent with the understanding of the factors resulting from the separation of the questions into groups according to the questionnaire. In other words, the goal of confirmatory factor analysis is to determine whether the data fit a hypothetical measurement model. Each analysis related to a specific number of underlying dimensions (from 3 to 6), since the criterion for selecting the sub-datasets depended on the existence or absence of dimensions such as supporting teaching, according to the curriculum.

Next, a series of exploratory factor analyses were also carried out to test whether the collected data corresponded to a latent factor variable structure similar to the one arising from the separation of the questionnaire statements. In this way, the structure created by the questionnaire data alone was tested without any prior evaluation of the statements. Confirming the factorial structure of the questionnaire provides evidence for its validity and the significance of the results extracted from it.

The free and open-source statistical software "Jamovi", version 2.4 (https://www.jamovi.org/, accessed on 15 December 2023) [19] was utilized for conducting exploratory factor analysis (EFA) and confirmatory factor analysis (CFA) in this study. "Jamovi" provides a user-friendly graphical interface for applying many statistical and machine learning techniques using the R language. The software streamlined factor extraction, rotation, and model testing in an accessible workflow for the performed factor analyses.

## 3. Results

*3.1. Confirmatory Factor Analysis*

Firstly, the factor loadings of the variables were computed. Table 4 presents the results of this process for the dataset containing all factors (subset #1 in Table 3). For conciseness purposes, only CFA factor loadings for this subset are presented in this paper; the complete data for all subsets are available in the Supplementary Materials of this paper. By analyzing the results of the CFAs, we can observe that the factor loadings were strong (above 0.6), indicating a strong relationship between the variable and the factor in most variables, suggesting that the tool is a suitable representation of the underlying structure. Additionally, the *p*-values indicate that the factor loadings are statistically significant. In this case, all factor loadings are highly significant ($p < 0.001$) [20]. Similar results were obtained for the loadings in the other datasets.

**Table 4.** CFA factor loadings; dataset contains all factors.

| Factor | Variable | Estimate | SE | Z | *p*-Value |
|---|---|---|---|---|---|
| | var1 | 1.075 | 0.0145 | 74.2 | <0.0001 |
| | var2 | 1.083 | 0.0140 | 77.1 | <0.0001 |
| | var3 | 1.162 | 0.0152 | 76.5 | <0.0001 |
| | var4 | 1.152 | 0.0152 | 75.8 | <0.0001 |
| | var5 | 0.938 | 0.0174 | 54.0 | <0.0001 |
| | var6 | 1.015 | 0.0158 | 64.1 | <0.0001 |
| Course | var7 | 0.867 | 0.0186 | 46.7 | <0.0001 |
| | var8 | 0.862 | 0.0168 | 51.2 | <0.0001 |
| | var9 | 0.855 | 0.0170 | 50.1 | <0.0001 |
| | var10 | 0.442 | 0.0164 | 26.9 | <0.0001 |
| | var13 | 0.689 | 0.0167 | 41.3 | <0.0001 |
| | var14 | 0.964 | 0.0162 | 59.4 | <0.0001 |
| Supporting teaching (if applicable) | var11 | 0.596 | 0.0233 | 25.6 | <0.0001 |
| | var12 | 0.882 | 0.0225 | 39.2 | <0.0001 |

**Table 4.** *Cont.*

| Factor | Variable | Estimate | SE | Z | *p*-Value |
|---|---|---|---|---|---|
| Assignments (if applicable) | var15 | 1.016 | 0.0154 | 66.0 | <0.0001 |
| | var16 | 1.026 | 0.0154 | 66.8 | <0.0001 |
| | var17 | 0.975 | 0.0183 | 53.3 | <0.0001 |
| | var18 | 1.271 | 0.0158 | 80.5 | <0.0001 |
| | var19 | 1.290 | 0.0158 | 81.7 | <0.0001 |
| | var20 | 1.176 | 0.0181 | 64.8 | <0.0001 |
| | var21 | 1.183 | 0.0162 | 73.1 | <0.0001 |
| Teaching staff | var22 | 1.220 | 0.0153 | 79.8 | <0.0001 |
| | var23 | 1.274 | 0.016 | 79.6 | <0.0001 |
| | var24 | 1.247 | 0.0152 | 81.9 | <0.0001 |
| | var25 | 1.165 | 0.0155 | 75.1 | <0.0001 |
| | var26 | 1.071 | 0.0154 | 69.7 | <0.0001 |
| | var27 | 1.172 | 0.0156 | 75.0 | <0.0001 |
| Laboratory (if applicable) | var28 | 0.984 | 0.0196 | 50.1 | <0.0001 |
| | var29 | 0.750 | 0.0177 | 42.5 | <0.0001 |
| | var30 | 1.260 | 0.0158 | 79.6 | <0.0001 |
| | var31 | 1.282 | 0.0159 | 80.8 | <0.0001 |
| | var32 | 1.057 | 0.0184 | 57.3 | <0.0001 |
| Student self-assessment | var33 | 0.842 | 0.0152 | 55.4 | <0.0001 |
| | var34 | 0.778 | 0.0171 | 45.4 | <0.0001 |
| | var35 | 0.812 | 0.0146 | 55.8 | <0.0001 |
| | var36 | 0.891 | 0.0163 | 54.7 | <0.0001 |
| | var37 | 0.735 | 0.0215 | 34.2 | <0.0001 |

On the other hand, the results indicated a less-than-adequate fit for all datasets based on several fit indices. The $\chi^2$ test was significant (*p*-values < 0.001) for all datasets, indicating a statistically significant divergence between the hypothesized model (as indicated by the questionnaire) and the observed data. Recognizing the disadvantages of the $\chi^2$ test, such as the sensitivity to sample size, with larger samples leading to smaller *p*-values, we also used additional fit measures such as Tucker's TLI [21], SRMR [22], and RMSEA [23] (Table 5). The values for the comparative fit index (CFI) [24] ranged from 0.859 to 0.894, falling below the recommended threshold of ≥0.95 for good fit [25]. The Tucker–Lewis index (TLI) also failed to reach adequacy across datasets, ranging from 0.845–0.884, with values not meeting the ≥0.95 threshold. The standardized root mean square residual (SRMR) fell within the recommended range of ≤0.08 for most of the models (0.0518–0.0757), indicating adequate absolute fit on this index only. However, the root mean square error of approximation (RMSEA) exceeded the recommended cutoff of ≤0.06 for a good fit [26,27], with values ranging from 0.0809 to 0.0896. The 90% confidence intervals for RMSEA also confirmed poor fitting across all datasets.

**Table 5.** CFA, goodness of fit indexes.

| | $\chi^2$ | N | *p*-Value | CFI | TLI | SRMR | RMSEA | RMSEA CI 90% Lower | RMSEA CI 90% Upper |
|---|---|---|---|---|---|---|---|---|---|
| No Lab-No AS/NT-No TS | 13,627 | 7395 | <0.001 | 0.876 | 0.863 | 0.0554 | 0.0855 | 0.0882 | 0.0867 |
| No Lab-No AS/NT-TS | 8787 | 4033 | <0.001 | 0.874 | 0.860 | 0.0528 | 0.0848 | 0.0832 | 0.0863 |
| No Lab-AS/NT-No TS | 31,551 | 9725 | <0.001 | 0.862 | 0.850 | 0.0564 | 0.0896 | 0.0888 | 0.0904 |
| No Lab-AS/NT-TS | 54,430 | 485 | <0.001 | 0.894 | 0.884 | 0.0518 | 0.0835 | 0.0829 | 0.0841 |
| Lab-No AS/NT-No TS | 5741 | 2259 | <0.001 | 0.859 | 0.845 | 0.0757 | 0.0833 | 0.0814 | 0.0852 |
| Lab-No AS/NT–TS | 2961 | 395 | <0.001 | 0.861 | 0.847 | 0.0690 | 0.0828 | 0.0800 | 0.0856 |
| Lab-AS/NT-No TS | 13,014 | 3464 | <0.001 | 0.868 | 0.857 | 0.0671 | 0.0809 | 0.0797 | 0.0821 |
| Lab-AS/NT-TS | 18,509 | 4274 | <0.001 | 0.886 | 0.877 | 0.0575 | 0.0826 | 0.0816 | 0.0836 |
| Total | | 48,008 | | | | | | | |

No Lab-No AS/NT-No TS: Datasets from courses without Laboratory, Assignment, Supporting Teaching.
Lab-AS/NT-TS: Datasets from courses with Laboratory, Assignment, Supporting Teaching.

Taken together, these results show inadequate model fit, suggesting that the theoretical model did not fit the empirical data well. Modifications to the hypothesized model are needed to improve fit across datasets before the interpretation of the findings. This was evident in all subsets of data and in almost all goodness-of-fit measures studied. Although the cutoff criteria for CFA are not absolute [27], the fact that there is almost total agreement between the indexes in all data subsets confirms this claim.

*3.2. Exploratory Factor Analysis*

The next stage was to perform a series of exploratory factor analyses (EFAs), comparing the factors present in the questionnaire structure with those extracted from the data extension. Two distinct approaches were employed to this end. The first method was to extract factors from the data extension without any constraint on the number of factors to be identified. The second method was to extract factors from the data extension, requesting that the variables within each analyzed subset be grouped to a number of factors equal to that of the subset being analyzed. These two analyses are presented in the following paragraphs.

As noted above, the first approach employed in this stage was to extract the dimensions of the questionnaire as derived from the data, performing a series of exploratory factor analyses (EFAs) without imposing any prerequisites concerning the grouping of the variables. All subsets of data were analyzed in the same way. Assuming that the resulting factors are correlated with each other, the promax rotation method in conjunction with the maximum likelihood extraction method was used. The expected factor structure, according to the questionnaire for the dataset with all dimensions, is shown in Table 6 below.

**Table 6.** Theoretical association between variables and dimensions.

| Dimensions | Questions |
| --- | --- |
| Factor 1 (Course) | var1, var2, var3, var4, var5, var6, var7, var8, var9, var10, var13, var14 |
| Factor 2 (Supporting teaching) | var11, var12 |
| Factor 3 (Assignments) | var15, var16, var17, var18, var19, var21 |
| Factor 4 (Teaching staff) | var22, var23, var24, var25, var26, var27, var28 |
| Factor 5 (Lab) | var29, var30, var31, var32 |
| Factor 6 (Student) | var33, var34, var35, var36, var37 |

The results of all analyses showed that, based on the eigenvalue criterion bigger than 1, the extracted factors were only two instead of three to six, as suggested in the questionnaire (Course, Supporting teaching, Assignments, Teaching staff, Lab, Self-assessment), with no particular conceptual similarity. An example from the subset of data that does not include support teaching, assignments, and labs is shown in Table 7, where the expected structure based on the questionnaire is not confirmed. The same pattern is repeated in all datasets.

According to the literature, the Keiser criterion [28] of eigenvalues greater than one is not absolute [29]; eigenvalue limits less than one and between 0.5 and 0.8 were tested. In these cases, too, the expected number of dimensions was not obtained. Failure to identify a number of dimensions equal to what is expected in each dataset indicates that a mismatch exists between the intended conceptual model and the one that actually emerges in the data.

The second approach followed in this EFA was to perform a factor analysis with a specific number of factors, corresponding to the number of dimensions of the questionnaire for each subset of the data. It was found that the variables that, according to the questionnaire, corresponded to (i) the courses (var1 to var10) and (ii) the teaching staff (var22 to var28) were actually associated with one factor, in a quite distorted fashion compared to the original conceptual grouping of the questionnaire (an example is shown in Table 8). In contrast, the questions related to students' self-assessment (var33–var37) appeared to relate to the same factor, distinct from the one to which courses and teaching staff relate. Clustering other variables into dimensions was also found to be biased compared to the theoretical model.

**Table 7.** EFA Factor Loadings, No Lab-No AS/NT-No TS dataset. Based on eigenvalues > 1.

| | Factor | | |
|---|---|---|---|
| | **1** | **2** | **Uniqueness** |
| var1 | 0.873 | | 0.289 |
| var2 | 0.877 | | 0.328 |
| var3 | 0.935 | | 0.261 |
| var4 | 0.901 | | 0.278 |
| var5 | 0.503 | | 0.709 |
| var6 | 0.732 | | 0.433 |
| var7 | 0.318 | | 0.854 |
| var8 | 0.339 | | 0.806 |
| var9 | 0.327 | | 0.731 |
| var10 | | 0.166 | 0.98 |
| var13 | 0.139 | | 0.948 |
| var14 | 0.573 | | 0.64 |
| var22 | 0.962 | | 0.195 |
| var23 | 0.868 | | 0.229 |
| var24 | 0.906 | | 0.241 |
| var25 | 0.828 | | 0.364 |
| var26 | 0.748 | | 0.485 |
| var27 | 0.782 | | 0.393 |
| var28 | 0.612 | | 0.59 |
| var33 | | 0.579 | 0.659 |
| var34 | | 0.556 | 0.517 |
| var35 | | 0.657 | 0.533 |
| var36 | | 0.809 | 0.364 |
| var37 | | 0.589 | 0.686 |

**Table 8.** EFA, factorial structure, No Lab-No AS/NT-No TS dataset, based on number of factors expected (3).

| Variable | Factor | | | |
|---|---|---|---|---|
| | **1** | **2** | **3** | **Uniqueness** |
| var1 | 0.812 | | | 0.29 |
| var2 | 0.809 | | | 0.328 |
| var3 | 0.899 | | | 0.262 |
| var4 | 0.816 | | | 0.273 |
| var5 | 0.359 | | | 0.644 |
| var6 | 0.613 | | | 0.404 |
| var7 | 0.162 | | | 0.767 |
| var8 | 0.139 | | | 0.665 |
| var9 | | | 0.642 | 0.496 |
| var10 | | | 0.369 | 0.899 |
| var13 | | | 0.273 | 0.909 |
| var14 | 0.493 | | | 0.632 |
| var22 | 0.942 | | | 0.191 |
| var23 | 0.876 | | | 0.209 |
| var24 | 0.927 | | | 0.213 |
| var25 | 0.884 | | | 0.311 |
| var26 | 0.74 | | | 0.479 |
| var27 | 0.846 | | | 0.33 |
| var28 | 0.571 | | | 0.591 |
| var33 | | 0.629 | | 0.619 |
| var34 | | 0.545 | | 0.517 |
| var35 | | 0.678 | | 0.509 |
| var36 | | 0.805 | | 0.348 |
| var37 | | 0.536 | | 0.692 |

The allocation of the courses and teaching staff to the same factor, combined with the specific number of factors to be extracted, has been also reported in other studies [15–18]. Overall, extracting the expected number of dimensions resulted in an unexpected structure for all data subsets, which was not expected.

From a statistical perspective, the previous finding is associated with the high correlation coefficients between the variables related to the course (var1 to 5) and the teachers (var22 to 28), as shown in Figure 1 with italics. In particular, data show moderate to strong

average correlation coefficients within the factors' statements in most cases. The coefficient of correlation between each factor's questions is high, indicating that the questions meet the requirement of creating a factor. On the other hand, a strong correlation between factors' statements was found, even if the requirement for a proper structure is a smaller correlation coefficient. The strongest average correlations were seen:

1.  Between course factors and teaching staff factors (avg 0.76), leading to matching to a factor.
2.  Between course and assignment factors (avg 0.67).

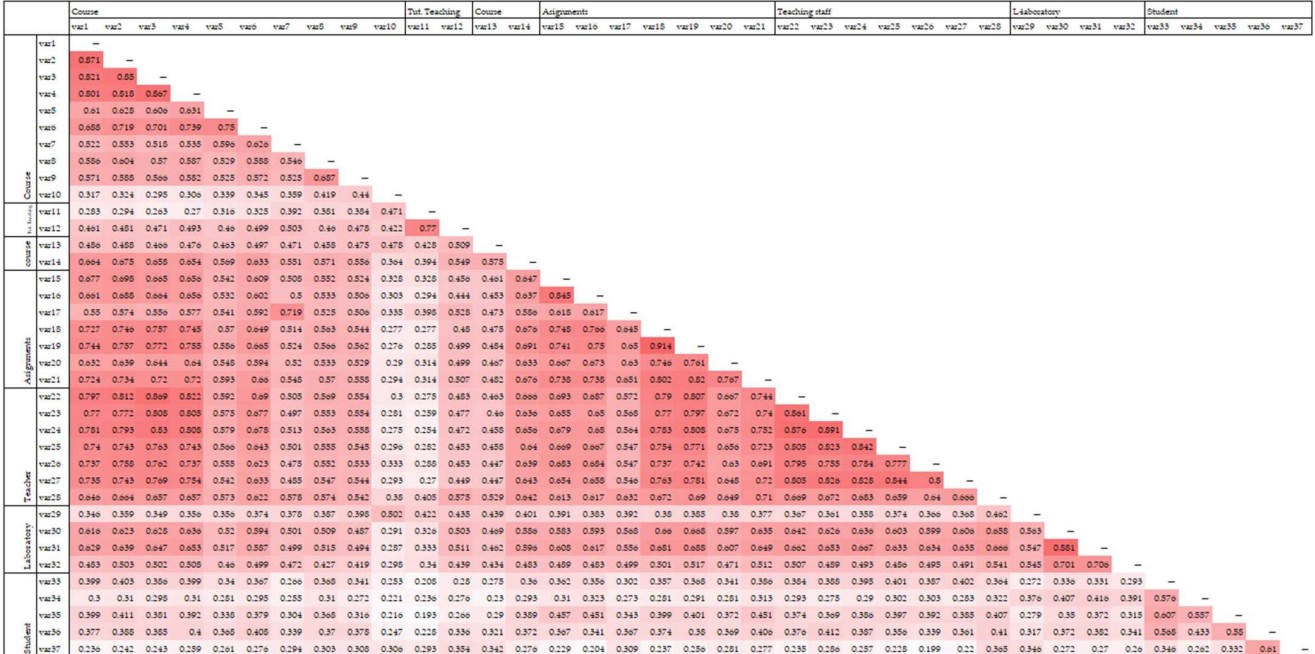

**Figure 1.** Correlations between items.

The high correlation between course and teaching staff factors is the root cause that led to a different factor structure than expected based on the under-study tool.

Correlations between supporting teaching and other factors are low or moderate, ranging from 0.19 to 0.57. Correlations between assignment and teaching staff factors are also strong, around 0.69, while the correlations between the assignment factor and the student factor are weak (avg 0.34). Teaching staff and laboratory factors show a moderate correlation of 0.53. Finally, student factors and other groups show low average correlations, ranging from 0.27 to 0.34. In summary, the strongest inter-factor correlations on average are seen between the factors of (i) course, (ii) teaching staff, and (iii) assignment. The weakest average correlations are between the student factor and the other factors.

The goodness-of-fit test indexes of EFAs with a specific number of factors indicate an acceptable or less than the good fit across all sub-datasets. The $p$-values for the $\chi^2$ test are all less than 0.0001, suggesting poor fit, but the $\chi^2$ test is affected by sample size. The corrected $\chi^2$ statistic based on the degrees of freedom shows an acceptable fit for "Lab-No AS/NT-TS" and a marginally acceptable fit for the "Lab-No AS/NT-No TS" subset.

Related to mean square error (RMSEA) values, we can observe that data subsets "Lab-AS/NT-TS" and "No Lab-AS/NT-TS" have the lowest values (0.711 and 0.741, respectively), while the "No Lab-AS/NT-No TS" data subset exhibits the highest value (0.0840). Considering that the RMSEA values of 0.05 and 0.08 correspond to "good fit" and "mediocre fit" [20], we can conclude that the determined RMSEA indicates a model fit that is marginally acceptable to mediocre. The 90% confidence intervals of RMSEA are narrow, demonstrating precision around the point estimates, below the 0.1 threshold. The Tucker–Lewis index (TLI) values exceed 0.90 for the "Lab-No AS/NT-TS" and "No Lab-AS/NT-TS" datasets, signifying

a good fit relative to the conventional cutoffs. The values of TLI for the rest of the subsets are between 0.857 and 0.895, indicating an adequate fit.

Regarding comparisons between datasets, the "Lab-AS/NT-TS" model exhibits the best fit based on the combination of a low RMSEA value (0.0711) and a high TLI (0.917). The "No Lab-No AS/NT-No TS" dataset also shows a good fit (TLI 0.892, RMSEA 0.0772, RMSEA 90% CI (0.0734, 0.0812)). In contrast, the "Lab-No AS/NT-TS" dataset has a lower TLI (0.848) compared to alternatives. Overall, the results provide some support for an acceptable fit across the sub-datasets, with some variability across them. However, the resulting factor structure is different from the expected one.

Finally, variables were clustered based on the factor loadings determined by EFA, and this clustering was compared to the clustering expressed by the questionnaire structure using the adjusted rand score index [30]. The computed adjusted rand score ranges from 0.2316 to 0.5128, indicating that the groupings observed in practice differ significantly from the theoretical ones.

The goodness-of-fit measures of exploratory factor analysis (EFA) without prerequisites of factors across the different datasets reveal varying degrees of model fit. The Root Mean Square Error of Approximation (RMSEA) generally suggests an acceptable-to-mediocre fit, with values ranging from approximately 0.0930 to 0.112; the Tucker–Lewis Index (TLI) values range from about 0.776 to 0.841, indicating an adequate fit for datasets. Despite significant chi-square ($\chi^2$) values, likely influenced by large sample sizes, the $\chi^2/df$ ratios generally suggest a reasonable fit in some datasets, ranging from 3.83 to 152.23. Additionally, all datasets exhibit $p$-values below 0.0001, indicating rejection of the null hypothesis of perfect fit. Overall, the EFA models show significant differences between observed and hypothesized structures. Also, the study of multiple fit indexes shows a moderate fit. Again, variables were clustered based on the factor loadings determined by EFA, and this clustering was compared to the clustering expressed by the questionnaire structure using the adjusted rand score index [30]. The computed adjusted rand score ranges from 0.0549 to 0.3670 (Table 9), indicating that the groupings observed in practice differ significantly from the theoretical ones. This range is lower than that observed when a specific number of outputs is requested to be extracted (Table 10).

**Table 9.** EFA, goodness-of-fit indexes, based on eigenvalues.

| | RMSEA | RMSEA 90% CI Lower | RMSEA 90% CI Upper | TLI | BIC | $\chi^2$ | df | $\chi^2/df$ | *p*-Value | Adjusted Rand Score |
|---|---|---|---|---|---|---|---|---|---|---|
| Lab-AS/NT-No TS | 0.0989 | 0.0961 | 0.102 | 0.809 | 753 | 4002 | 493 | 8.12 | <0.0001 | 0.1831 |
| Lab-No AS/NT-TS | 0.0977 | 0.0927 | 0.103 | 0.776 | −580 | 1435 | 348 | 4.12 | <0.0001 | 0.0659 |
| Lab-No AS/NT-No TS | 0.0979 | 0.924 | 0.104 | 0.785 | 594 | 1246 | 323 | 3.86 | <0.0001 | 0.162 |
| Lab-AS/NT-TS | 0.100 | 0.099 | 0.102 | 0.834 | 12,556 | 17,010 | 558 | 30.48 | <0.0001 | 0.122 |
| No Lab-AS/NT-TS | 0.112 | 0.111 | 0.113 | 0.799 | 68,100 | 72,462 | 476 | 152.23 | <0.0001 | 0.0549 |
| No Lab-AS/NT-No TS | 0.105 | 0.103 | 0.107 | 0.820 | 5061 | 7879 | 376 | 20.95 | <0.0001 | 0.1664 |
| No Lab-No AS/NT-TS | 0.107 | 0.104 | 0.109 | 0.790 | 2715 | 4704 | 65 | 72.37 | <0.0001 | 0.1067 |
| No Lab-No AS/NT-No TS | 0.093 | 0.089 | 0.097 | 0.841 | 522 | 2086 | 229 | 9.1 | <0.0001 | 0.367 |

**Table 10.** EFA, goodness-of-fit indexes, specific number of factors.

| | RMSEA | RMSEA 90% CI Lower | RMSEA 90% CI Upper | TLI | BIC | $\chi^2$ | df | $\chi^2/df$ | *p*-Value | Adjusted Rand Score |
|---|---|---|---|---|---|---|---|---|---|---|
| Lab-AS/NT-No TS | 0.0817 | 0.0787 | 0.0849 | 0.869 | −314 | 2520 | 430 | 5.86 | <0.0001 | 0.2504 |
| Lab-No AS/NT-TS | 0.0803 | 0.0746 | 0.0864 | 0.848 | −790 | 918 | 271 | 3.38 | <0.0001 | 0.2316 |
| Lab-No AS/NT-No TS | 0.0782 | 0.0719 | 0.0849 | 0.862 | 781 | 768 | 272 | 2.82 | <0.0001 | 0.4234 |
| Lab-AS/NT-TS | 0.0711 | 0.0696 | 0.0725 | 0.917 | 3579 | 7243 | 459 | 15.78 | <0.0001 | 0.3272 |
| No Lab-AS/NT-TS | 0.0741 | 0.0733 | 0.0749 | 0.912 | 22,185 | 25,700 | 373 | 68.9 | <0.0001 | 0.5128 |
| No Lab-AS/NT-No TS | 0.084 | 0.0819 | 0.0862 | 0.88 | 2149 | 4749 | 347 | 13.69 | <0.0001 | 0.2189 |
| No Lab-No AS/NT -TS | 0.0797 | 0.0768 | 0.0827 | 0.881 | 629 | 2277 | 227 | 10.03 | <0.0001 | 0.4578 |
| No Lab-No AS/NT-No TS | 0.0772 | 0.0734 | 0.0812 | 0.892 | −63.1 | 1351 | 207 | 6.53 | <0.0001 | 0.3537 |

*3.3. CFA and EFAs' Results*

Both confirmatory and exploratory factor analyses conclude a lack of agreement between the hypothesized structure of the research instrument capturing students' views. Based on the abovementioned findings, questions can be raised regarding the ability of students to distinguish the degree of satisfaction they receive from the course or the teacher, a view also adopted in [15]. An alternative (or complementary) interpretation is to attribute the findings to the lack of validity of the questionnaire. Validity refers to the assessment of whether a tool actually measures what it claims to measure or whether there is a systematic error. When we examine the structure and layout of the questionnaire, we note its face validity, as the statement questions are located under the factors that they are intended to relate to [29,31]. However, the structural and layout aspects are only a superficial assessment of the measurement tool in terms of the underlying dimensions, providing only a rough initial estimate of whether the content of the question–statements is conceptually relevant to the intended conceptual structure. In contrast, other forms of validity, such as internal and content validity, require the implementation of quantitative and qualitative methods to assess them. The need to use additional quantitative and qualitative tools has been underscored in other studies [17,18].

## 4. Discussion

This study highlighted the weaknesses of an important tool for evaluating teaching work and providing feedback from students, using an extensive set of data. Student evaluations of teaching (SETs) are commonly used in higher education as a measure of teaching effectiveness [8–19]. The present study is the first that evaluates the factorial structure of the specific questionnaire used in the University of the Peloponnese and other Greek universities. Confirmatory factor analyses were initially conducted to determine if the data matched the hypothesized factor structure implied by the grouping of statements into categories on the questionnaire. The results indicated a mediocre-to-poor model fit across all datasets, based on $\chi^2$, CFI, TLI, SRMR, and RMSEA values. This suggests that the data did not align well with the proposed factor structure.

Several exploratory factor analyses were performed in order to let the factors emerge directly from the data. From these analyses, only two or three factors were extracted, rather than the expected number of factors based on the questionnaire categories. Even when the theoretically expected number of factors was defined, the item loadings did not match the dimensions of the questionnaire. In particular, course evaluation and teacher evaluation variables consistently loaded onto the same factor. These findings are in line with other studies and evaluation tools [15–19,31].

The high positive correlation between course and teacher evaluation items explains their clustering on one factor. The average correlation between course and teaching staff factors was 0.76, representing the highest inter-factor correlation. This raises questions about the ability of students to distinguish the satisfaction they received from the courses and the instructors. It may also point to issues with the discriminatory power of the tool. Students may not be able to accurately assess the satisfaction they receive, and this may be due to the wording of the questions. Face validity of the tool seems reasonable, but a more rigorous examination of content and construct validity is required.

The EFA results lend some support to a low-to-acceptable fit when extracting the expected number of factors but with a different structure than expected. The RMSEA and TLI values were marginally adequate for most datasets. This indicates that the data may have an underlying factor structure that does not match the questionnaire structure.

Overall, CFA and EFA findings converge to suggest limitations of this evaluation tool, while similar findings exist in the literature. The lack of validity in the questionnaire has important consequences. The study shows difficulties in reaching precise conclusions about some facets of teaching, which limits the tool's usefulness as a means of assessing teaching factors. In order to improve the usefulness of the questionnaire for collecting information

from students and evaluating teaching effectiveness in general, future research should focus on improving and revalidating it.

Modifications are needed to improve the model fit. Adding, removing, or revising items to increase validity between the course and teaching staff factors is also suggested, as the high correlation between them indicates that students currently do not adequately distinguish these dimensions.

Restructuring the questionnaire sections and items, as in supporting teaching, could also help. Reducing the correlations between factors by removing redundant elements or differentiating the content could help delineate the factors. Examining the validity of factor scores based on variables such as grades or demographic characteristics of students and teachers may reveal other issues. Periodic re-validation, following improvements, will ensure that the factor structure will continue to fit new students and curricula over time. Finally, improving content validity through experts is also important. This collaborative approach not only strengthens content validity but also ensures the reliability and effectiveness of the tool.

Through these steps, the validity of the tool can be enhanced to better capture the key dimensions in student evaluations of teaching. Further reliability and validity testing on other datasets is recommended in order to improve the results obtained from these data. The examination of the effect of reducing statements on the reliability of the tool and the results obtained from it is another future field of study, as the current findings provide a first evaluation of the properties of this important student feedback tool.

## 5. Conclusions

The present study aimed to evaluate the validity of a student evaluation of teaching (SET) questionnaire utilized at the University of the Peloponnese. The analysis covered 48.008 records for eight academic years and for all university courses. These records spanned across 68 undergraduate and postgraduate programs and 2380 courses. The study indicated an insufficient fit to the theorized underlying structure. However, the existing literature emphasizes the necessity of empirically testing the instruments used to ensure that they provide unbiased data to inform administrative and academic decision-making.

The lack of validity of the assessment tool highlights the importance of empirical evaluation of these instruments to ensure that they provide accurate and objective feedback. This study is useful for institutions worldwide that use student feedback for administrative and academic decision-making, as it highlights the need to validate the instruments in order to produce useful data. The findings may have implications for methods of improving the assessment tools used worldwide to more accurately assess teaching quality and collect insightful feedback from students.

The lack of validity of the questionnaire holds important implications at both theoretical and practical levels. Theoretically, refinement and revalidation of the questionnaire were highlighted as necessities for adapting the data generated to the anticipated theoretical framework. Practically, the study identified difficulties drawing specific conclusions pertaining to diverse facets of instruction, hampering its utility as an evaluation tool for teaching quality and highlighting the need for enhancement. Future research could explore refining and revalidating the questionnaire to bolster its usefulness for obtaining student feedback and assessing teaching effectiveness.

**Supplementary Materials:** The following supporting information can be downloaded at: https://www.mdpi.com/article/10.3390/higheredu3020013/s1.

**Author Contributions:** Conceptualization, C.V.; methodology, I.P.; software, I.P.; validation, C.V., M.W. and A.K.; formal analysis, I.P.; investigation, I.P.; resources, I.P.; data curation, I.P.; writing—original draft preparation, I.P.; writing—review and editing, C.V. and M.W.; visualization, I.P.; supervision, A.K.; project administration, C.V. All authors have read and agreed to the published version of the manuscript.

**Funding:** This research received no external funding.

**Institutional Review Board Statement:** The study has been approved by the board of Research Ethics and the Integrity Committee of the University of Peloponnese-Peloponnese (Decision number 2675/12 February 2024).

**Informed Consent Statement:** Not applicable.

**Data Availability Statement:** Restrictions apply to the availability of these data. Data were obtained from University of the Peloponnese (permission of use decision: 11600/21 November 2023) and can be made available from the institution, subject to the approval of a relevant request.

**Conflicts of Interest:** The authors declare no conflicts of interest.

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
