# Peer review of "On the Quality and Validity of Course Evaluation Questionnaires Used in Tertiary Education in Greece"

_2813-4346, doi:10.3390/higheredu3020013_

Round 1

Reviewer 1 Report

Comments and Suggestions for Authors

The research in the article is interesting and applicable in other European universities, so I recommend continuing with this field of study.

Author Response

Dear Reviewer, 

Thank you very much for your warm comments.

Ilias Papadogiannis

Reviewer 2 Report

Comments and Suggestions for Authors

The chosen topic of the paper is quite modern and has elements of novelty. The paper is relevant to the topic, reasonably clear, and well-organized. It also conforms to the parameters and scope of the journal.

2. The majority of the references are recent, relevant publications that have been published in the last ten years, however, there are some sources that are quite old and may be substituted by newer ones.

3. The manuscript's outcomes and discoveries unequivocally demonstrate the paper's originality.

4. The research results are proven, and the data are presented in graphs and tables. Every table is precisely correct. They accurately depict the facts and are also simple to read and understand. The statistical analysis (p-value) used in the study supports the validity of the findings.

5. Information regarding the global impact of the research can be included in the Conclusion section.

Comments on the Quality of English Language

In general, the language of the article is correct, clear and corresponds to the academic style, but there are rare cases of the use of more informal constructions that should be replaced, such as "in relation to", "with respect to" etc. Also, some other minor grammar corrections are required, like using the articles and gerund constructions.

Author Response

Dear reviewer,
Thank you for your kind comments and valuable feedback on my paper.

Regarding comment 2. (literature references beyond the decade). 9 changes were made. In my opinion, there will be no need for changes to some of the basic texts such as [24]
Comment 5: Another paragraph was added to the conclusions.

Thanks again for your comments

Ilias Papadogiannis

Reviewer 3 Report

Comments and Suggestions for Authors

I believe this is important foundational work on the value of student course evaluations. Clearly there are issues with this particular tool. Your data analysis is appropriately robust, and your findings are compelling. I hope you continue with this line of inquiry to inform true upgrades to the existing tool. 

Author Response

Dear reviewer,

I truly would like to thank you for your comments.

Ilias Papadogiannis